# Mitophagy in Cell Death Regulation: Insights into Mechanisms and Disease Implications

**DOI:** 10.3390/biom14101270

**Published:** 2024-10-09

**Authors:** Jiani Lin, Xinyao Chen, Yuyang Du, Jiapeng Li, Tingting Guo, Sai Luo

**Affiliations:** The 1st Affiliated Hospital of Harbin Medical University, No. 23, Youzheng Street, Nangang District, Harbin 150000, China; 2019183110@hrbmu.edu.cn (J.L.); 814895@hrbmu.edu.cn (X.C.); 814850@hrbmu.edu.cn (Y.D.); hydljp@hrbmu.edu.cn (J.L.); 2023021109@hrbmu.edu.cn (T.G.)

**Keywords:** mitophagy, mitochondria, intracellular homeostasis, cell death

## Abstract

Mitophagy, a selective form of autophagy, plays a crucial role in maintaining optimal mitochondrial populations, normal function, and intracellular homeostasis by monitoring and removing damaged or excess mitochondria. Furthermore, mitophagy promotes mitochondrial degradation via the lysosomal pathway, and not only eliminates damaged mitochondria but also regulates programmed cell death-associated genes, thus preventing cell death. The interaction between mitophagy and various forms of cell death has recently gained increasing attention in relation to the pathogenesis of clinical diseases, such as cancers and osteoarthritis, neurodegenerative, cardiovascular, and renal diseases. However, despite the abundant literature on this subject, there is a lack of understanding regarding the interaction between mitophagy and cell death. In this review, we discuss the main pathways of mitophagy, those related to cell death mechanisms (including apoptosis, ferroptosis, and pyroptosis), and the relationship between mitophagy and cell death uncovered in recent years. Our study offers potential directions for therapeutic intervention and disease diagnosis, and contributes to understanding the molecular mechanism of mitophagy.

## 1. Introduction

Mitochondrial homeostasis is achieved by maintaining a balance between mitochondrial biogenesis and the removal of damaged mitochondria. This balance is essential for maintaining normal cellular functions and mass homeostasis. Under reactive oxygen species (ROS) stress, nutrient deficiency, and cellular aging, intracellular mitochondria are depolarized, leading to damage. To ensure mitochondrial network stability and intracellular environment maintenance, cells use autophagy to selectively endocytose and degrade dysfunctional or damaged mitochondria, which is called mitophagy [1,2]. Mitochondrial oxidative stress and ROS production are closely related to mitophagy [3]. Normally, mitophagy regulates programmed cell death-associated genes as a stress response mechanism. Mitophagy is induced by a variety of pathological processes such as oxidative stress and inflammation, and plays a key role in preventing cell death. This association suggests a general mechanism via which mitophagy is related to cell death [4]. However, despite the extensive literature on this topic, the understanding regarding the crosstalk between mitophagy and cell death is lacking. Therefore, this review reviews current research on the mechanism of mitophagy, main cell death mechanisms, and the association between mitophagy and cell death.

## 2. Mitophagy

Since the discovery of mitophagy, its mechanisms of action have garnered considerable attention. Two types of mechanisms have been proposed: ubiquitin- and receptor-mediated pathways [5]. In the following sections, we discuss these two mitophagy mechanisms (Figure 1, Table 1).

### 2.1. Ubiquitin-Mediated

The ubiquitin-dependent mitophagy pathway is mainly mediated by the ubiquitination of mitochondrial surface proteins. Among these, the most extensively studied is the PTEN-induced putative kinase 1 (PINK1)–Parkin pathway [6]. The PINK1–Parkin pathway, a key mechanism of mitophagy, is present in various human tissues. It plays a major role in mediating mitophagy in skeletal muscle, cardiac muscle, brain, T-lymphocytes, erythrocytes, lung, and tumor cells [7]. PINK1 is a highly conserved mitochondrial protein that is underexpressed in normal mitochondria because of its constant transportation to the intermembrane space for proteolytic cleavage, followed by transport to the cytoplasm and, finally, proteasomal degradation [8]. Upon mitochondrial membrane potential (MMP, ΔΨm) disruption, the pathway of PINK1 into the inner mitochondrial membrane (IMM) is blocked. This leads to steady PINK1 accumulation in the outer mitochondrial membrane (OMM), resulting in the recruitment of Parkin to the damaged mitochondria [9]. Next, the spatial conformation of Parkin changes, exposing the catalytic cysteine. This converts cysteine to an activated E3 ubiquitin ligase. Parkin then acts as an E3 ubiquitin ligase that binds ubiquitin molecules to the substrate protein, ultimately leading to the recognition and degradation of ubiquitin molecules by the ubiquitin-labeled proteasome [10]. Following translocation, PINK1 directly phosphorylates the ubiquitin-like structural domain, Ser65, thereby increasing the E3 activity of Parkin [11]. Upon localization to the mitochondria, Parkin cooperates with PINK1 to amplify the initial signal by modifying the mitochondria at the Ub chain, which is subsequently phosphorylated by PINK1. This process leads to the ubiquitination and degradation of several mitochondrial dynamics-associated proteins, such as Miro. As a result, mitochondrial transport is suspended, and damaged mitochondria are separated from the healthy mitochondrial network [12]. Recruitment of the mitophagy receptor proteins, OPTN, NDP52, and NBR1, to ubiquitinated mitochondria occurs simultaneously via their ubiquitin-binding domains. Subsequently, light chain 3 (LC3) is recruited to damaged mitochondria via the large body protein double FYV-containing protein 1 (DFCP1), allowing OPTN to initiate autophagosomes via LC3 interaction region (LIR)-mediated initiation [13,14]. Autophagosomal membrane formation around ubiquitinated mitochondria depends on the binding of NDP52 and OPTN to the core autophagy proteins, FIP200 and ATG9A, respectively [13,15]. In addition to the PINK1–Parkin pathway, several ubiquitin-dependent pathways are Parkin-independent. In other words, PINK1 can also directly recruit the autophagy receptors, OPTN and NDP52, to the mitochondria via ubiquitin phosphorylation, thereby promoting mitophagy [16]. This suggests that OPTN and NDP52 exist independently of Parkin. However, Parkin amplifies the PINK1-induced signaling pathway and enhances mitophagy.

### 2.2. Receptor-Mediated

Unlike PINK1–Parkin-mediated mitophagy ubiquitination, many proteins contain LIR structural domains on the outer or inner mitochondrial membrane, which are mitophagy receptors. They directly bind to the essential autophagy protein, LC3, without ubiquitination, thereby initiating mitophagy and selectively destroying damaged mitochondria [17]. In mammals, these receptors include FUN14 structural domain-containing 1 (FUNDC1), BCL2-interacting protein 3 (BNIP3)/Nip3-like protein X (NIX), and other receptors. In the following section, we describe receptor-mediated mitophagy.

#### 2.2.1. FUNDC1

FUNDC1 is an OMM protein containing cytoplasm-exposed LIR at its N-terminus [18]. FUNDC1 is normally phosphorylated by Src kinases and casein kinases 2 (CK2) at the Tyr-18 and Ser13 sites, respectively. The human FUNDC1 protein is widely expressed in various tissues, with a notable presence in the heart [19]. Under mitochondrial stress, FUNDC1 is dephosphorylated by the phosphatase PGAM5 or other phosphatases, leading to enhanced FUNDC1-LC3 interactions and subsequent selective mitophagy [20]. In contrast, although FUNDC1-mediated mitophagy is Parkin-independent, another mitochondrial E3 ubiquitin ligase, MARCH 5, can regulate mitophagy under hypoxic conditions via ubiquitin degradation of FUNDC1 [21]. In summary, FUNDC1-induced mitophagy is regulated by both ubiquitin degradation and phosphorylation.

In addition, FUNDC1 directly interacts with dynamic mitochondrial proteins, such as the mitochondrial fission and endosomal fusion proteins, DRP1 and OPA1, respectively [22]. Hypoxia reduces the ability of FUNDC1 to interact with OPA1 and increases its association with DRP1, inducing mitochondrial breakage and selective mitochondrial elimination [22,23]. Thus, FUNDC1 induces mitophagy via mitochondrial dynamic regulation.

FUNDC1-dependent mitophagy is crucial for maintaining mitochondrial mass and function in platelets, thereby supporting platelet activation, ultimately impacting blood oxygen levels and the risk of myocardial infarction [24]. Interestingly, hypoxic preconditioning has been shown to induce excessive FUNDC1-dependent mitophagy, preserving mitochondrial integrity, reducing oxidative stress, suppressing inflammation, and effectively protecting against I/R-induced cardiac injury [25,26]. Additionally, FUNDC1 plays a vital role in preserving normal mitochondrial morphology and function in cardiomyocytes, while its deficiency has been implicated in the development of metabolic disorders [27]. Recent research has linked the ablation of FUNDC1 to worsened obesity and insulin resistance from high-fat diets, demonstrating that FUNDC1 deficiency in white adipose tissue can impair mitophagy and mitochondrial function, leading to increased oxidative stress and inflammation [28]. Notably, Fu et al. found that FUNDC1-mediated mitophagy deficiency in skeletal muscle can counteract chronic high-fat diet-induced obesity by enhancing insulin sensitivity and glucose homeostasis, illustrating the interplay between muscle and adipose tissues in promoting adipose metabolism [29]. These discoveries underscore the pivotal role of FUNDC1-mediated mitophagy in systemic metabolism regulation.

#### 2.2.2. BNIP3 and NIX

BNIP3, located in the OMM, belongs to a subfamily of the anti-apoptotic B-cell lymphoma-2 (BCL-2) family, with a BH3 structural domain that is potentially capable of binding to LC3 [29]. Ser17 and Ser24 phosphorylation in the LIR motif promotes its interaction with LC3B and GATE16 [30]. Furthermore, ULK1 is responsible for phosphorylating BNIP3 at Ser17, leading to increased BNIP3 stability c-Jun N-terminal kinase (Jnk)1/2-mediated BNIP3 phosphorylation at Ser60, and Thr-66 regulates protein stability and prevents its proteasomal degradation during hypoxia [31,32]. The homolog of BNIP3, NIX (or BNIP3L), shares considerable sequence homology with BNIP3 and can induce mitophagy by directly binding to LC3 via the BH3 structural domain. NIX phosphorylation at Ser34 and Ser35 enhances its ability to bind to LC3B and recruit autophagosomes to the mitochondria [33]. Cd-treated cells to promote mitophagy exhibited increased Ser81 phosphorylation, which has been shown to be required for nix-mediated mitophagy [34]. Taken together, these results indicate that the function and stability of BNIP3 and NIX are regulated by phosphorylation.

NIX’s significance in mitophagy was initially shown in the development of mature erythrocytes, with the upregulation of NIX expression observed in terminally differentiated erythrocytes [35,36]. The process of nix-mediated mitophagy plays a crucial role in mitochondrial clearance, facilitating the transition from reticulocytes to mature erythrocytes. Additionally, it was found that high levels of BNIP3 expression were able to restore mitochondrial clearance in NIX-deficient reticulocytes. Notably, NIX-mediated mitophagy has been identified as essential for the differentiation of retinal ganglion cells and the successful reprogramming of somatic cells into induced pluripotent stem cells [37,38]. Koentjoro et al. found that pharmacological induction of NIX restored mitophagy and mitochondrial function in a cell line derived from a patient with Parkinson-related PD [39]. This study suggests an interaction between different mitophagy mechanisms. Unlike Parkin, BNIP3 or NIX do not appear to play a crucial role in the elimination of depolarized mitochondria from cells; however, some data suggest that they can ameliorate Parkin-mediated mitophagy and compensate for the absence of functional Parkin [40]. The co-action of BNIP3 with Parkin has been validated in models of skeletal and cardiac myocyte senescence, muscular dystrophy, type 2 diabetes, and other diseases [41,42,43]. Pharmacological PINK1 activation has been shown to ameliorate pathology in Parkinson’s disease models.

## 3. Cell Death

Various forms of cell death have been identified, each of which relies on a different subset of proteins to activate and execute their respective pathways [44]. In recent years, a progressively deep and comprehensive understanding of cell death patterns has emerged to describe the complex crosstalk mechanism among different pathways. Among these, three well-characterized pathways—apoptosis, ferroptosis, and pyroptosis—have been studied regarding their impact on cell fate, owing to their interactions with mitophagy. The following section focuses on the mechanisms underlying these three forms of cell death.

### 3.1. Apoptosis

Apoptosis is a highly conserved “suicide” program in metazoan cells [45]. An essential feature of apoptosis is mitochondria-mediated cytochrome c release, which is regulated by the balance between pro- and anti-apoptotic proteins of the BCL-2 family, initiating caspases (caspase-8, -9, and -10), and effector caspases (caspase-3, -6, and -7) [46]. Two main pathways are responsible for mitochondrial cytochrome c release downstream of the caspase activation cascade that triggers apoptosis, namely the intrinsic and extrinsic pathways. ROS play a crucial role in mediating lipid peroxidation and promoting apoptosis. Under oxidative stress, reactive intermediates are generated, which oxidize polyunsaturated fatty acids (PUFAs) in the membrane lipid bilayer. This process leads to the formation of aldehydes [47]. The resulting lipid peroxidation products interact with membrane receptors, transcription factors, and inhibitors. This interaction stimulates the activation of both endogenous and exogenous apoptotic signaling pathways [48]. A detailed discussion of these apoptotic pathways and the role of ROS will follow.

Oligomerization of the BCL-2 family proteins, BAK and BAX, activates intrinsic pathways [49]. BAK/BAX oligomers form pores in the OMM, leading to the release of cytochrome c into the cytoplasm [50]. BAK/BAX activation is regulated by pro-apoptotic and anti-apoptotic BCL-2 family proteins, such as BAD and BID, and BCL-2, respectively [51]. Additionally, reactive oxygen species (ROS) disrupt the mitochondrial membrane by oxidizing polyunsaturated fatty acids (PUFAs), which irreversibly opens the mitochondrial permeability transition pore. This process results in compromised mitochondrial membrane permeability and increased release of cytochrome c. Cytochrome c binds to Apaf-1, recruits procaspase-9, and forms apoptotic bodies [52,53,54]. During apoptosis, caspase-9 is activated by the hydrolytic cleavage of its own protein, initiating a caspase-processing cascade [55,56]. The exogenous pathway is in turn activated by membrane receptors such as tumor necrosis factor (TNF) receptor 1 (TNFR1), death receptors, or Toll-like receptors (TLRs) [57,58]. These proteins induce the formation of death-inducing signaling complexes (DISCs) via TNFR1-related death structural domain (TRADD) proteins, FAS-related death structural domain (FADD) proteins, receptor-interacting serine/threonine protein kinase 1 (RIPK1), and procaspase-8, among others [59,60]. Lipid peroxidation and its byproducts, particularly 4-hydroxynonenal (4-HNE), serve as significant signaling molecules that promote gene expression and trigger apoptosis through the Fas pathway, with effects that depend on concentration and duration [61]. Inhibitors of apoptosis (cIAPs) ubiquitinate RIPK1, stabilize the complex, and induce nuclear factor kappa B (NF-κB) transcription [62]. FADD-like IL1-β-converting enzyme-inhibitory protein (FLIP) is also present in DISC and limits caspase-8 activity while promoting cell survival, cell proliferation, and pro-inflammatory cytokine production [63]. Imbalances in this pathway, such as those imposed by cellular stress, allow caspase-8 and caspase-10 activation, which in turn trigger the caspase activation cascade. Once activated, executioner caspases (caspase-2, -6, -8, and -10) lead to programmed apoptosis [64]. Apoptotic cells release messengers in the form of nucleosomal structures, shedding receptors, anti-inflammatory metabolites, and molecules encapsulated in apoptotic extracellular vesicles (ApoEVs) [65]. Phosphatidylserine (PS) molecules are exposed on the outer surface of the plasma membrane and provide phagocytes with an “eat-me” signal [66]. This process involves a series of events contributing to the orderly elimination of damaged or unwanted cells from the body.

### 3.2. Ferroptosis

Ferroptosis is a form of cell death caused by glutathione peroxidase 4 (GPX4) inactivation and iron accumulation, in turn leading to lipid ROS accumulation. There are several major metabolic pathways involved, as follows:

#### 3.2.1. Iron Metabolism

Extracellular trivalent iron ions bind to the transferrin receptor (TFR1) on the cytosolic membrane via transferrin (TF), initiating endocytosis and endocytic vesicle formation [67]. Trivalent iron is reduced to divalent iron by a six-transmembrane epithelial antigen of prostate 3 (STEAP3) and is then transported intracellularly via divalent metal transport protein (DMT1) or binds to ferritin to form an iron pool [68,69]. In the iron pool, ferritin can be coated by autophagic lysosomes mediated by nuclear receptor coactivator protein 4 (NCOA4), which in turn degrades and releases large amounts of Fe^2+^ [70]. When ferroptosis occurs, a large amount of free Fe^2+^ accumulates in the cell. Notably, free Fe^2+^ is highly oxidative and easily reacts with H_2_O_2_ in the Fenton reaction, generating hydroxyl radicals that cause oxidative damage to DNA, proteins, and membrane lipids and promote lipid peroxidation to damage cell membranes, ultimately leading to cell death [71,72].

#### 3.2.2. Lipid Peroxidation

Lipid peroxidation describes a reaction where lipids lose hydrogen atoms under the action of free radicals or lipid peroxidases, leading to oxidation, breakage, and shortening of the lipid carbon chain, as well as the production of cytotoxic substances, which ultimately cause cellular damage [73].

In ferroptosis, the deleterious effects of lipid peroxidation are mainly observed in the oxidative degradation of two key biofilm components: polyunsaturated fatty acids (PUFAs) and phosphatidylethanolamine (PE). PUFAs are the major components of phospholipids in cell and organelle membranes and are important substrates for PE synthesis. PE, in turn, is the main component of the interior of the phospholipid bilayer, which is the structural basis for maintaining cell membranes’ fluidity and plays a key role in PE regulation [74]. Owing to their high affinity for free radicals, PUFAs are susceptible to oxidation [75]. Lipid peroxidation alters the molecular conformation of PUFA, thereby disrupting the fluidity and stability of the cell membrane structure, increasing membrane permeability, and rendering cells susceptible to rupture and death [76]. In contrast, PE does not exhibit a high affinity for free radicals. Consequently, PE forms PE-AA/AA/AA/AdA with adrenoyl-CoA (AdA-CoA), which leads to susceptibility of oxidation to free radicals or arachidonic acid 15-lipoxygenase (ALOX15), resulting in the production of the cytotoxic lipohydroperoxide PE-AA/AA-OOH. This process promotes ferroptosis [77,78].

#### 3.2.3. Glutathione (GSH) Metabolism

GSH, a water-soluble tripeptide comprised of the amino acid residues glutamic acid, cysteine, and glycine, plays a crucial role as an antioxidant in the human body. Reduced GSH reduces H_2_O_2_ to H_2_O, scavenges free radicals, and maintains the intracellular free radical equilibrium, as well as acts as a co-factor of glutathione peroxidase 4 (GPX4), participating in lipid peroxide (LOOH) reduction to repair biofilms and prevent the onset of ferroptosis [79,80]. The process of executing ferroptosis requires both GSH depletion and GPX4 enzyme activity inhibition. Cells synthesize GSH to maintain intracellular levels mainly via the extracellular uptake of cysteine and glutamate via the cysteine/glutamate transporter protein system Xc-, which consists of solute carrier family 7 member 1 (SLC3A2) and solute carrier family 3 member 2 (SLC7A11). When system Xc- subunit activity is inhibited, cellular cysteine uptake is insufficient, which in turn blocks GSH synthesis and is rapidly depleted by H_2_O_2_ and lipid peroxidation [73,81].

Current studies on the three regulatory mechanisms of ferroptosis have not shown a clear interaction among lipid metabolism, iron metabolism, and amino acid metabolism. These mechanisms tend to function independently.

### 3.3. Pyroptosis

Pyroptosis is a type of programmed cell death characterized by rapid plasma membrane disruption, followed by the release of cellular contents and pro-inflammatory mediators (cytokines), including IL-1β and IL-18. Pyroptosis is induced by gasdermins, a family of transmembrane pore-forming proteins that are activated by inflammasome-dependent or non-inflammatory vesicle-dependent pathways. Inflammatory vesicles are cytoplasmic immune signaling complexes that cause inflammation and pyroptosis and aggregate according to pathogen- or injury-related molecular patterns (PAMPs or DAMPs) [46,82]. Inflammatory vesicles typically include sensors, adapters, and the zymogen procaspase-1 (zymogen) [83]. Inflammatory vesicle-initiating sensors are pattern recognition receptors (PRRs), which include nucleotide-binding oligomerized structural domains (NODs) and leucine-rich repeat sequence (LRR) receptors (NLRs) [84]. Upon detection of a specific stimulus, the sensor recruits inflammatory vesicle adapter apoptosis-associated speck-like proteins containing caspase activation and recruitment domains (CARDs) (ASCs) to form multimeric complexes called “specks” [85]. The CARD of ASC allows the ASC to couple upstream sensor PRRs to the effector cysteine protease caspase-1, which then converts the effector into its biologically active form [86]. Activated caspase-1 and caspase-4/5/11 triggers pyroptosis via typical and atypical pathways, respectively [87].

In a typical pyroptosis pathway, PRRs (such as NLRs, AIM2, or pyrin) sense PAMPs and DAMPs in response to stimuli, which marks the beginning of functional inflammatory vesicle formation [86]. Upon recognizing pathogenic stimuli, PRRs bind to pro-CASP1 with the help of ASC speckles, culminating in inflammatory vesicle formation [88]. This results in CASP1 activation, whereafter active CASP1 cleaves the pyroptosis actuator GSDMD and releases the pore-forming structural domains from the inhibitory effect of GSDMD-c. In addition, active CASP1 matures pro-IL-1β and pro-IL-18 into active forms via protein hydrolysis [89]. Subsequently, the released GSDMD-N is recruited to the inner side of the cell membrane and oligomerizes to form transmembrane pores. The transmembrane pores lead to K+ spillover and water influx, resulting in cell swelling, cell rupture, and cellular content release, including of bioactive IL-1β and IL-18. This further amplifies the host inflammatory response [90,91].

In the atypical pathway, lipopolysaccharide, a component of the Gram-negative bacterial cell wall, is recognized by the host CASP4/5/11. Subsequently, the activated CASP4/5/11 initiates GSDMD lysis and pyroptosis [92,93]. Although CASP11 can cleave GSDMD, it cannot convert pro-IL-1β and pro-IL-18 into their biologically active forms [94]. Thus, the atypical pathway requires caspase-1 to produce mature IL-1β and IL-18 [92].

Cellular senescence is closely linked to the interplay between mitophagy and pyroptosis. Senescence is characterized by dysregulated mitophagy, leading to the accumulation of dysfunctional mitochondria, which serve as a crucial factor in driving various aspects of the senescence phenotype [95]. The dysfunctional mitochondria present in senescent cells release various damage-associated molecular patterns (DAMPs), particularly reactive oxygen species (ROS) and mitochondrial DNA (mtDNA) fragments. These DAMPs are recognized by the NLRP inflammasome, resulting in the assembly of inflammatory vesicles and activation of pro-inflammatory cytokines such as IL-1β and IL-18 [96]. Consequently, senescent cells become more vulnerable to programmed necrosis, termed scorched death, due to the dysregulation of mitophagy. Further investigation is necessary to elucidate the signaling pathway underpinning the relationship between senescence and mitophagy.

### 3.4. NETosis

Activated neutrophils can release nuclear DNA into the extracellular environment through a process known as NETosis, which captures and neutralizes pathogens [97]. The cellular mechanisms underlying NETosis are still being elucidated. Notably, the primary signals that initiate NETosis, distinct from other neutrophil responses, remain unidentified. However, NET release is initiated by the activation of surface receptors, leading to changes in intracellular calcium concentration, activation of kinase signaling cascades, and the production of ROS. These signaling events result in morphological changes, including increased cell spreading and altered cell shape. Research by Fuchs et al. indicates that ROS may not induce NETosis through the activation of surface receptors [98]. Traditionally, ROS production has been viewed as a direct mechanism for pathogen elimination via oxidative damage [99]. However, ROS also play a crucial role in the signaling pathways involved in NETosis. The two primary sources of ROS in neutrophils are NADPH oxidase and mitochondria. While the exact mechanism by which NADPH oxidase facilitates NET release is not fully understood, mitochondria have dual roles in this process. They mediate ROS production and can release mtDNA into the extracellular space. Studies by Douda et al. have demonstrated that both calcium ion carrier-induced NETosis and spontaneous NETosis in low-density granulocytes from systemic lupus erythematosus (SLE) patients require an increase in mitochondrial ROS production [100,101]. Therefore, during NETosis, ROS can be released from both mitochondria and NADPH oxidase. This raises the question of the role of mitophagy, a physiological process that scavenges ROS, in the onset and regression of NETosis. This question will be addressed through careful analyses in subsequent sections.

## 4. Mitophagy and Cell Death Crosstalk

The crosstalk between mitophagy and apoptosis is mediated by members of the BCL2 family and their interacting proteins, as well as other mitophagy-related proteins. Mitophagy selectively removes damaged or excess mitochondria. With severe injury, damaged mitochondria are not removed efficiently and the mitochondrial outer membrane permeability (MOMP) is altered, leading to cytochrome c translocation into the cytoplasm, which in turn activates caspases and triggering apoptosis [102]. MOMP is regulated by the BCL-2 family of proteins, including the anti-apoptotic BCL-2 and BCL-xL, pro-apoptotic BAX and BAK, and BH3-only proteins [103,104]. BCL-2/BCL-xL inhibits MOMP by preventing BAX/BAK activation by sequestering BH3-only proteins or by preventing oligopore formation in the mitochondrial outer membrane by sequestering activated BAX/BAK [105].

BCL2 regulates autophagy via interaction with the BH3-only structural domain of the autophagy regulator, BECN1, thereby blocking autophagosome formation and binding to BECN1 [104,106,107,108]. The relative amounts of intracellular BECLIN1 and Bcl-2 (or other Bcl-2 family members) complexed with each other determine the threshold for cell homeostasis to cell death [108,109]. Some BCL2-interacting proteins, such as FUNDC1 and BNIP3, disrupt the interaction between BECN1 and the anti-apoptotic BCL2 family and positively regulate mitophagy (Figure 2).

The interaction between FUNDC1 and BCL-xL is controlled by the multimerization of PGAM5, which exists in equilibrium between the dimeric and multimeric states and acts as a molecular switch to sense different levels of stress and generate appropriate responses [110,111,112]. In unstressed cells, BCL-xL is unphosphorylated and binds to the mitochondrial phosphatase, PGAM5, whereas FUNDC1 is inhibited by phosphorylation [112]. Mild stress, such as nitrite-induced oxidative stress, switches PGAM5 to a multimeric state, which then fails to bind and reactivate BCL-xL to prevent mitochondrial damage. Multimeric PGAM5 binds and dephosphorylates FUNDC1 to activate mitochondrial fission and subsequent mitophagy, thereby eliminating damaged mitochondria and promoting cell survival [110,111]. Severe stress such as vincristine-induced mitotic arrest is sensed by kinases that phosphorylate BCL-xL to reduce its interaction with BAX and BAK, thereby reducing its anti-apoptotic function. In its dimerized state, PGAM5 binds and dephosphorylates BCL-xL at Ser62 to reactivate its antiapoptotic function. Lethal stress, usually a combination of oxidative stress and mitotic arrest, promotes cell death by inactivating BCL-xL and blocking FUNDC1-dependent mitophagy [111].

Under normoxic conditions, BECLIN1 forms low-affinity complexes with Bcl-XL and Bcl-2 via its atypical BH3 structural domain, thereby reducing the autophagy rate. Under hypoxic conditions, the oxygen gradient between the vasculature and necrotic regions is reduced, and the atypical BH3 structural domains of the two HIF target genes, BNIP3 and BNIP3L, compete with the BECLIN1-bcl-2 and BECLIN1-bcl-xl complexes, releasing BECLIN1 and thus enhancing autophagy [113,114,115,116]. Hypoxia-inducible factor-1α (HIF-1α)-BNIP3-mediated mitophagy in renal tubular cells attenuates apoptosis and senescence during acute kidney injury (AKI) [107].

BCL-xL also inhibits PINK1–Parkin-dependent mitophagy by inhibiting Parkin translocation to depolarized mitochondria in two regulatory ways. BCL-xL binds directly to Parkin in the cytoplasm to prevent its translocation from the cytoplasm to mitochondria [117]. However, it also binds directly to PINK1 on the mitochondria to interfere with the stable recruitment of Parkin to the mitochondria [105,118]. This interaction has been demonstrated in various disease models such as cancer, diabetes, and osteoporosis [119,120,121,122,123]. Moreover, they have great potential for application in cancer treatments. For example, ketoconazole induces apoptosis in cancer cells via COX-2 downregulation and PINK1–Parkin-mediated mitophagy activation [119].

Ferroptosis and cellular metabolism are closely linked [124,125,126]. In terms of physiological processes, intracellular redox homeostasis is imbalanced during ferroptosis. The levels of antioxidants, such as GSH and GPX4, decrease, and the levels of oxidants, such as divalent iron ions and ROS, increase [127,128]. Mitochondria produce adenosine triphosphate (ATP) via oxidative phosphorylation to maintain redox and calcium homeostasis. However, because of its role as the main organelle in hemoglobin and iron–sulfur cluster production, it contains large amounts of mitochondrial iron and integrates iron metabolism within the cytoplasm [129]. The binding of iron to mitochondrial ferritin prevents ROS production, whereas mutation or degradation of mitochondrial ferritin can lead to mitochondrial iron overload [130]. Thus, an important relationship exists between the homeostasis of mitochondrial function, mass, and ferroptosis. This was confirmed by a striking morphological feature of ferroptotic cells observed under electron microscopy: ferroptotic cells typically contain mitochondria that shrink with increasing membrane density [131]. Ferroptosis leads to cellular damage, and mitophagy plays a protective role by inhibiting ROS release from dysfunctional mitochondria.

Using a mitophagy protocol to deplete mitochondria in cells increases or inhibits the onset of ferroptosis but does not affect other forms of necrotic cell death, such as necrosis [131,132,133]. These results suggest that mitophagy, as a mechanism for selectively degrading damaged mitochondria via autophagic flux and for maintaining mitochondrial mass homeostasis and functional normalcy, is closely implicated in ferroptosis onset and outcome. These findings further imply that the effects of mitophagy on ferroptosis are complex and two-sided [134]. In the early stages of mild stress or iron overload, mitophagy may sequester iron in autophagosomes. Ferroptosis can lead to cellular damage, and mitophagy may play a protective role by inhibiting ROS release from dysfunctional mitochondria, thereby reducing ROS-derived material in ferroptosis [135]. However, extensive mitophagy may ultimately provide additional iron, which amplifies lipid peroxidation and ferroptosis. For example, biphasic changes in O-GlcNAcylation, a major nutrient sensor of glucose flux, regulate ferroptosis by coordinating ferritin autophagy and mitophagy after receiving a ferroptosis stimulus, such as RSL3 (commonly used ferroptosis inducer). Pharmacological or genetic inhibition of o-GlcN acylation promotes ferritin autophagy, leading to the accumulation of unstable iron in the mitochondria, whereas the o-GlcN acylation inhibition leads to mitochondrial breakage and enhanced mitophagy, providing an additional source of unstable iron and rendering cells sensitive to ferroptosis. Thus, the degree of autophagic flux during autophagy may vary, as may the effect of autophagy on ferroptosis (Figure 3).

This led us to speculate how the biphasic transition between mitophagy and ferroptosis occurs. Ferroptosis and mitophagy are mediated by several interacting network nodes that influence each other and play unique roles. The current findings focus on the association of the two with ferritin autophagy, as well as mitochondrial molecular and functional abnormalities.

Ferroptosis requires autophagy. Among the identified autophagic processes, NCOA4-promoted ferritinophagy, BECN1-mediated systemic Xc-inhibition, RAB7-dependent lipophagy, STAT3-induced lysosomal membrane permeabilization, SQSTM1-dependent clock autophagy, HSP90-associated chaperonin-mediated autophagy, and PINK1-associated mitophagy regulate cellular ferroptosis [136,137]. Ferritin autophagy plays a key role in ferroptosis, and interactions between mitophagy and ferritin autophagy pathways in turn promote ferroptosis. For example, in a study on protein O-GlcN acylation, ferroptosis induction was accompanied by O-GlcNAc transferase inactivation, leading to ferritin de-O-GlcN acylation and activation of NCOA4-mediated ferritin autophagy and mitophagy. Ferroptosis and ferritin autophagy collectively provided a source of unstable iron required for the Fenton reaction, leading to accelerated ROS generation and lipid peroxidation [68]. Moreover, simultaneous inhibition of ferritin autophagy and mitophagy (e.g., dual NCOA4 and PINK1 downregulation) almost completely blocked ferroptosis. This is further illustrated by Singh et al. on diabetic retinopathy (DR). High glucose levels induced thioredoxin-interacting protein (TXNIP) upregulation, and the associated redox stress leads to mitochondrial dysfunction, mitophagy, ferritin autophagy, lysosomal instability, and unstable iron reacting with H_2_O_2_ to form ∙OH, causing membrane phospholipid peroxidation via the Fenton reaction, which leads to ferroptosis [138].

Abnormalities in certain mitochondrial molecules and their functions can affect mitophagy and ferroptosis. The major mitophagy receptor protein FUNDC1 is strongly associated with ferroptosis. The absence of FUNDC1 prevents paraquat-induced ferroptosis and uncontrolled mitophagy and effectively attenuates cardiac injury [139]. In mice fed a short-term high-fat diet, FUNDC1 regulates ferroptosis via ACSL4 and affects disease course, leading to metabolic and cardiac remodeling and contractile dysfunction [140]. Voltage-dependent anion channels (VDACs) are important proteins involved in the mitophagy and ferroptosis crosstalk. VDAC is a target of Parkin-mediated ubiquitination and acts as a mitochondrial docking site, recruiting Parkin from the cytoplasm to defective mitochondria and inducing mitophagy [141,142,143]. VDAC, one of the most abundant OMM proteins, allows iron and metabolites, such as respiratory substrates ADP, Pi, and ATP, to cross the outer membrane, and the erastin-induced opening of VDAC-mediated mitochondrial iron uptake may accelerate ferroptosis [135,144]. In addition, the OMM protein, MitoNEET, also known as CISD1 (CDGSH iron-sulfur structural domain 1), is a redox-sensitive (2Fe-2S) cluster protein that is critical for iron perception and regulation, as well as ROS homeostasis [144,145]. Owing to its redox sensitivity, CISD oxidizes VDAC, closing the pores and potentially disrupting the flow of metabolites from VDAC [145,146]. Thus, VDAC inhibition leads to mitochondrial tricarboxylic acid (TCA) cycle arrest, which attenuates mitochondrial membrane potential hyperpolarization and lipid peroxide accumulation and triggers an increase in mitophagy-dependent ROS, leading to ferroptosis [124,146,147]. Furthermore, knockdown of the mitochondrial localization protein, CISD3, substantially accelerates lipid peroxidation and exacerbates free iron accumulation triggered by system Xc suppression or cystine deprivation, which in turn promotes cellular ferroptosis [146,148].

Pyroptosis is a type of inflammatory cell death. As cytoplasmic immune signaling complexes that cause inflammation and pyroptosis, inflammasomes are central to the crosstalk between mitophagy and pyroptosis. In this section, we briefly discuss the molecular mechanisms by which inflammasome vesicles affect the mitophagy and cell death crosstalk (Figure 4).

Mitophagy dysregulation leads to mitochondrial dysfunction. When mitochondria are damaged, their components are recognized by cytoplasmic pattern recognition receptors (members of the nod-like receptor family of NLRP3 inflammatory vesicles) as DAMPs when released into the cytoplasm [149,150]. NLRP3 inflammatory vesicles inactivate pre-caspase-1 into active caspase-1 and cleave pro-inflammatory IL-1β into mature IL-1β, leading to inflammation and premature cell death. To counteract the damaging effects of mtROS and inflammatory vesicles, mitophagy requires the removal of dysfunctional mitochondria [6,151]. Moreover, a member of the Nod-like receptor (NLR) family, which also has a mitochondrial-targeting sequence, contains an LIR, and its LIR motif binds to LC3 to induce mitophagy [6,152]. Thus, balanced activation of the inflammasome–mitophagy pathway may contribute to protective host immunity and the prevention of deleterious inflammatory responses, thereby maintaining human health. However, dysregulation of the inflammasome and mitophagy pathways may lead to pathological inflammatory responses that further exacerbate mitochondrial damage, resulting in pyroptosis [153].

Innate immune components are primarily involved in the association between inflammatory vesicles and mitophagy to maintain immune homeostasis. NF-κB activates NLRP3-inflammatory vesicles by inducing pro-IL-1β and NLRP3 expression, which is key to NLRP3 inflammatory vesicle activation [154]. Simultaneously, NF-κB prevents excessive inflammation and exerts its anti-inflammatory activity by inducing a delayed accumulation of the autophagy receptor p62/SQSTM1. Stimulation of NLRP3 activation triggers caspase-1- and NLRP3-independent mitochondrial injury and leads to the direct release of NLRP3 inflammasome activators, including mtDNA and mtROS [155]. Damaged mitochondria undergo Parkin-dependent ubiquitin binding and are specifically recognized by p62, which induces mitochondrial autophagic clearance [156]. Thus, the NF-κB-p62-mitophagy pathway is necessary to control NLRP3 inflammasome hyperactivation and IL-1β-mediated inflammation, an intrinsic system for tissue repair and anti-inflammatory homeostasis [154].

In addition to NF-kB, other factors are involved in cell death and mitophagy crosstalk. Inflammatory vesicles activate caspase-1. CASP1 inhibits mitophagy and amplifies mitochondrial damage, which is mediated in part by the cleavage of Parkin, a key mitophagy regulator. Only cells with CASPASE1-mediated mitochondrial damage exhibit low levels of FSC/SSC, consistent with cell death, and CASPASE1-mediated kinetics of mitochondrial injury precede the appearance of the low-profile FSC/SSC. Thus, inactivation of mitophagy may be an early event after inflammatory vesicle activation. These findings suggest that mitophagy eliminates pyroptosis-induced mitochondrial damage by interacting with NLRP3 and CASP1 as a compensatory mechanism for pyroptosis-induced cell death [157].

Recent papers have proposed that the cGAS-STING pathway is one of the major pathways for immune defense against various types of pathogens. Under certain cellular stress conditions, endogenous genomic and mitochondrial DNA can enter the cytoplasm, further activate cyclic guanosine-adenylate synthase (cGAS), and trigger immune and inflammatory responses [158]. cGAS induces the production of type I interferon (IFN-I) by recognizing cytoplasmic DNA molecules and catalyzing the generation of the second messenger, cGAMP (cyclic GMP-AMP), which binds to interferon gene-stimulating factor (STING). This mechanism highlights the critical role of the cGAS-STING pathway in activating immune responses upon detection of intracellular DNA [159]. Abnormal DNA accumulation plays a crucial role in facilitating the sustained activation of the cGAS-STING pathway, leading to the continuous release of inflammatory cytokines. These cytokines are instrumental in various physiological processes such as autoimmunity, sterile inflammatory responses, and cellular senescence [160]. The reaction of mtROS with newly synthesized mtDNA to generate oxidized mtDNA and the induction of mitochondrial dysfunction by NLRP3 secondary signaling activators such as ATP lead to the release of oxidized mtDNA into the cytoplasm and activation of NLRP3 inflammasomes, triggering inflammasome assembly and activation [161,162,163]. Furthermore, oxidative stress contributes to the generation of oxidized mtDNA, which in turn triggers the activation of the NLRP3 inflammasome and promotes inflammation [163]. Both Zhou et al. and Nakahira et al. established that mtROS are necessary for NLRP3 activation. Additionally, Nakahira et al. demonstrated that the release of mtDNA is crucial for NLRP3 activation and that this process relies on ROS production. In the context of acute liver injury, ROS production induces hepatocyte pyroptosis by activating the NLRP3/caspase-1/GSDMD signaling pathway and promoting the extracellular release of mtDNA [164]. It was found that GSDMD is capable of inducing rapid mitochondrial collapse, resulting in the accumulation of mtDNA in the cytoplasm. This accumulation of mtDNA promotes its release from the cell upon plasma membrane rupture [165]. Moreover, the inhibition of mitophagy due to hepatocyte-specific FUNDC1 ablation was observed to lead to the accumulation of dysfunctional mitochondria, which subsequently triggers the release of cytoplasmic mtDNA and the activation of caspase-1 [166]. Therefore, the activation of mtDNA/cGAS/STING signaling by hindering mitophagy may contribute to the promotion of cellular pyroptosis. This newly discovered signaling pathway offers a promising avenue for further investigation into the interaction between mitophagy and pyroptosis.

### NETosis

The molecular and cellular connections between NETosis and mitophagy remain largely unexplored [167]. Current research identifies two primary aspects of their relationship: first, NETosis necessitates an increase in mitochondrial ROS production; second, mitophagy mitigates the accumulation of defective mitochondria, thereby reducing ROS levels. Chu et al. demonstrated that inhibiting NETosis formation reduces ferroptosis in intestinal endothelial cells by enhancing FUNDC1-dependent mitophagy [168]. Activated neutrophils are essential for pathogen elimination through the release of neutrophil extracellular traps (NETs) [100]. However, excessive NET production can cause local tissue damage and may exert pro-inflammatory effects [169]. This overproduction of NETs impairs mitophagy, exacerbating mitochondrial dysfunction. Consequently, this dysfunction leads to a loss of mitochondrial membrane potential, increased cytosolic and mitochondrial ROS, and decreased mitochondrial ATP synthesis. Furthermore, Yazdani et al. highlight that NETs not only influence energy production but also play a role in maintaining mitochondrial homeostasis by regulating mitochondrial fission, fusion, and autophagy [170]. These findings raise critical questions, such as whether the overproduction of ROS due to excessive NETosis creates a positive feedback loop that enhances NETosis biogenesis. Additionally, is there a potential upstream–downstream relationship between mitophagy and NETosis? Further in-depth studies are essential to elucidate the role of mitophagy in NETosis.

## 5. Disease-Relevant Cases

In recent years, mechanistic studies exploring the crosstalk between mitophagy and cell death have enhanced our understanding of various disease models associated with multiple mitochondrial dysfunctions. For instance, the pivotal role of osteoblast apoptosis in age-related bone loss and osteoporosis has been extensively investigated. Studies have demonstrated that late-phase oxidized protein products (AOPPs) can prompt the generation of ROS, initiate mitochondria-dependent endogenous apoptotic pathways, and worsen oxidative stress conditions, ultimately accelerating osteoblast apoptosis [120,171]. Similarly, in diseases such as diabetes mellitus, acute kidney injury, neuronal injury, and amyotrophic lateral sclerosis, mitophagy inhibition stemming from the absence or reduced expression of mitophagy proteins like FUNDC1, PINK1, and BNIP3 has been shown to lead to various detrimental effects. These effects include triggering mitochondrial damage, arresting mitochondrial biosynthesis, inducing oxidative stress, causing accumulation of mitochondrial debris, and ultimately leading to apoptosis [27,113,172,173,174,175]. In their study, Zhou et al. focused on the role of casein kinase 2α in causing myocardial ischemia/reperfusion. They highlighted that by post-transcriptionally modifying the Ser13 locus, casein kinase 2α enhances the phosphorylation of FUNDC1, resulting in the inactivation of FUNDC1. This process effectively prevents mitophagy, without directly impacting the biosynthesis or expression of FUNDC1 [176]. This finding was further supported by Zheng et al.’s investigation into diabetic nephropathy [177].

Of interest is the crosstalk between hepatocytes and macrophages in metabolic dysfunction-associated steatohepatitis (MASH). Studies have pointed out that mitophagy in hepatocytes involves many regulatory factors and signaling pathways. Interestingly, the negative regulation of pyroptosis by mitophagy is pathway-independent. Impaired mitophagy can result from either BNIP3 transcriptional repression or blockade of ubiquitination of Parkin, leading to an increased release of pro-inflammatory and pro-fibrotic cytokines in macrophages. Specifically, in MASH, hepatocyte inflammatory vesicle activation induced by Parkin auto-ubiquitination and downstream blockade of outer mitochondrial membrane (OMM) ubiquitination stimulates macrophage production of TNF-α. This, in turn, further blocks mitophagy, creating a vicious feedback loop [178,179].

Mitochondrial damage plays a crucial role in causing neuronal death, while the process of mitochondrial quality control (MQC) is vital in preserving mitochondrial homeostasis to support neuronal survival. Impaired MQC intensifies ferroptosis by triggering excessive mitochondrial division and mitophagy [180]. The crosstalk between mitophagy and ferroptosis has garnered increased attention in neurological disorders, particularly neurodegenerative disorders resulting from neuronal injury. Shen et al. highlighted the pivotal role of NLRP6 as a mitophagy sensor in preserving mitochondrial homeostasis in hippocampal NSPCs. Their findings underscore the significant contribution of NLRP6 in this process [181]. In Parkinson’s disease, mitophagy mechanisms orchestrate the formation of memory T cells. Interleukin-15 (IL-15) triggers the upregulation of mitophagy regulators like Parkin and NIX. This upregulation is crucial as it counters ferroptosis and helps prevent metabolic dysfunction that can be induced by impaired mitophagy. Ultimately, IL-15 supports memory T cell formation by maintaining proper mitochondrial function through the regulation of mitophagy pathways [182]. Studies on ferroptosis and mitochondrial kinetic mechanisms, such as mitophagy, mitochondrial translocation, and mitochondrial fusion, have revealed that neuronal ferroptosis is regulated in an MQC-dependent manner [180].

The relationship between NETosis and mitophagy is significant for understanding tumor cell growth. Neutrophils, which constitute a substantial portion of inflammatory cells in the tumor microenvironment (TME) of various malignancies, play a crucial role in this context [100,183,184,185]. While neutrophils are primarily recognized for their role in antimicrobial defense, tumor-associated neutrophils (TANs) have been shown to promote tumor growth and metastasis at multiple stages of cancer progression. Recent studies indicate that NET-treated cancer cells activate the mitochondrial biogenesis-related gene PGC1a and enhance the expression of fission- and fusion-related proteins, including DRP-1, MFN-2, PINK1, and Parkin. This suggests that NETosis can directly modify the metabolic programming of cancer cells, thereby facilitating tumor growth [170]. Consequently, the emerging evidence supporting the pro-tumor functions of neutrophils through NETosis opens new avenues for therapeutic research targeting neutrophils in cancer treatment.

## 6. Targeted Therapy for Mitophagy

Various small-molecule compounds have been used in treating neurodegenerative, cardiovascular, and osteoarthritis diseases due to the observed crosstalk between mitophagy and cell death in different disease models [186,187].

In the field of biomedical research, various pharmacological interventions have been investigated for their potential in managing ischemic stroke and cardiovascular diseases. Notably, tissue-type plasminogen activator (tPA) is widely recognized as the primary thrombolytic drug for the clinical treatment of ischemic stroke due to its neuroprotective effects, which involve the modulation of FUNDC1-mediated mitophagy [188]. This mechanism underlines the importance of mitophagy in mitigating the consequences of ischemic stroke on brain tissue. Conversely, Carfilzomib has emerged as a promising therapeutic agent for alleviating ischemic brain injury by targeting BNIP3L degradation, as suggested in a study by Wu et al. Their findings highlighted the role of proteasomal degradation of BNIP3L and mitophagy dysregulation in cerebral ischemia, shedding light on a novel approach for intervention [189]. Moreover, in the context of cardiovascular diseases, certain medications such as empagliflozin, resveratrol, and berberine have shown efficacy in preserving the structural and functional integrity of cardiac microvasculature. These drugs achieve their effects through promoting mitophagy activation and suppressing apoptosis, thereby offering potential clinical benefits for patients with cardiovascular conditions. The elucidation of these mechanisms underscores the importance of targeting mitophagy and apoptosis pathways in the development of novel therapeutic strategies for ischemic stroke and cardiovascular diseases [190,191,192,193].

Mitophagy has been extensively investigated in osteoarthritis models, establishing a strong correlation with the mechanism of cell death. One notable example is the disease-modifying drug kd 025, known for enhancing mitophagy in chondrocytes. This drug has shown promising results in alleviating cartilage degeneration and reducing apoptosis [171]. Similarly, pharmacological activation of PINK1–Parkin-mediated mitophagy in AOPP-stimulated osteoblasts has shown promise in alleviating this pathogenesis [120,171]. Moreover, therapeutic modalities like Focused Low-Intensity Pulsed Ultrasound (FLIPUS) have been utilized to stimulate mitophagy effectively [194]. In the realm of cancer therapy, various drugs that target mitophagy, such as artesunate, triphenylphosphine ruthenium complexes, temozolomide, and levatinib, have demonstrated the ability to induce apoptosis in drug-resistant tumor cells [195,196,197,198]. Particularly in the treatment of hepatocellular carcinoma, mitophagy-targeted drugs are emerging as viable therapeutic approaches with great potential.

Mitophagy-targeting drugs have emerged as promising candidates for the treatment of bacterial infections and inflammation-related diseases due to the well-studied crosstalk between pyroptosis as a mode of inflammatory cell death and mitophagy. Luo et al. found that certain drugs with anti-inflammatory properties, such as bergamot lactone, can modulate NLRP3 inflammatory vesicle activation and cellular pyroptosis by enhancing mitophagy and preserving mitochondrial homeostasis [199]. This evidence underscores the potential therapeutic value of targeting mitophagy in managing chronic inflammatory diseases [164,200]. Among these, neuroinflammation has been studied extensively, particularly in hippocampal neurons. Drugs such as ginsenoside Rb1, quercetin, betulinic acid, and pine sapogenins work by modulating mitophagy to inhibit cellular pyroptosis, thereby helping to maintain nervous system homeostasis through inflammation suppression [201,202,203,204]. Recent research has explored the potential of targeting mitophagy to trigger tumor cell pyroptosis as a strategic approach in response to the heightened resistance of tumor cells to apoptosis-inducing anticancer medications. The concept of pyroptosis and the release of immunogenic mediators have been utilized in antitumor strategies to reprogram the tumor microenvironment. Studies have discovered that the activation of mitophagy in tumor cells, including those in breast cancer and glioblastoma, can induce cellular pyroptosis, leading to a potent anti-tumor immune reaction [205,206,207]. Various studies have shown the efficacy of targeting mitophagy blockade in enhancing cellular juxtaposition, especially within nanosystems. These nanosystems exploit a hybrid membrane composed of homologous cellular and mitochondrial membranes to enhance drug delivery. By combining the homing effect of cellular membranes with the directional fusion of subcellular membranes, these nanosystems deliver drug complexes loaded with mitophagy inhibitors like chloroquine to the mitochondria, ultimately promoting cellular juxtaposition. This innovative drug delivery system has demonstrated significant improvements in antitumor effects [206,207].

Certain ferroptosis inhibitors (such as antioxidants) affect mitophagy. Similarly, certain drugs associated with mitophagy regulation play important roles in ferroptosis. Oxidative stress activates ferroptosis and mitophagy during ischemia-reperfusion (I/R) injury following acute kidney transplantation. Ferroptosis may lead to kidney injury, whereas mitophagy may play a protective role by reducing the release of ROS from dysfunctional mitochondria [208]. In addition, WJ460, which targets the oncoprotein myoferlin, decreases the abundance of ferroptosis coregulators (xc-cystine/glutamate transporter protein and GPX-4). Myoferritin, which is associated with low survival in many cancer types, controls mitochondrial structure and respiration. Targeting myoferritin with WJ460 triggers mitophagy and ROS accumulation, ultimately leading to lipid peroxidation and ferroptosis [209]. The synergistic effect of these drugs is important for treating diseases and provides new research ideas for sensitivity studies of certain anticancer drugs.

## 7. Conclusions

Mitophagy, the primary mechanism for monitoring and removing damaged or excess mitochondria, plays a vital role in mitochondrial quality control and critically maintains optimal mitochondrial numbers, normal functions, and intracellular homeostasis. The crosstalk between mitophagy and various forms of cell death such as apoptosis, ferroptosis, and pyroptosis has a major impact on the onset and progression of many diseases [210]. Although the role of mitophagy in cell death has been extensively studied, there are still some important knowledge gaps to be filled. The first is the in-depth study of the fine regulation of the mechanism. Although we understand that some key molecules, such as PINK1 and Parkin, play a role in mitophagy and cell death, the specific signaling pathways still need to be explored. In particular, the differences in regulatory mechanisms under different cell types and stress conditions are not well defined. The second is the pathogenesis and outcome of diseases caused by the interaction between the two. At present, most of the studies based on disease models or treatment methods can only reach conclusions at the phenomenon level. In the future, we need to focus on specific molecular events and pathways. Another point that deserves our attention is the overall understanding of MQC. The interaction of mitophagy with other quality control mechanisms, such as mitochondrial fusion and fission, plays an important role in crosstalk with cell death [180]. This requires a comprehensive understanding. The filling of these knowledge gaps will help to further understand the complex relationship between mitophagy and cell death, thereby promoting the research and treatment of related diseases. It can be seen that drugs such as empagliflozin, as mentioned above, activate mitophagy to determine cell fate. This class of agents has been increasingly used to treat the chemical induction of disease [190]. In cancer treatment, in addition to drugs such as artesunate that inhibit tumor growth by inhibiting mitophagy, new drugs that reprogram the tumor microenvironment by regulating mitophagy to make cancer cells more sensitive to chemotherapeutic drugs have also been put into use in fields such as breast cancer [202,211]. In the future, further studies on the specific mechanisms and regulatory networks will provide new ideas for the treatment of tumors with high anticancer drug resistance. At the same time, non-traditional induction modalities such as ultrasound and nanosystems have also provided new horizons for models of degenerative diseases, inflammatory diseases, cancer, and aging [176,206,212]. This will require collaboration across disciplines and substantial preclinical studies and clinical trials.

## Figures and Tables

**Figure 1 biomolecules-14-01270-f001:**
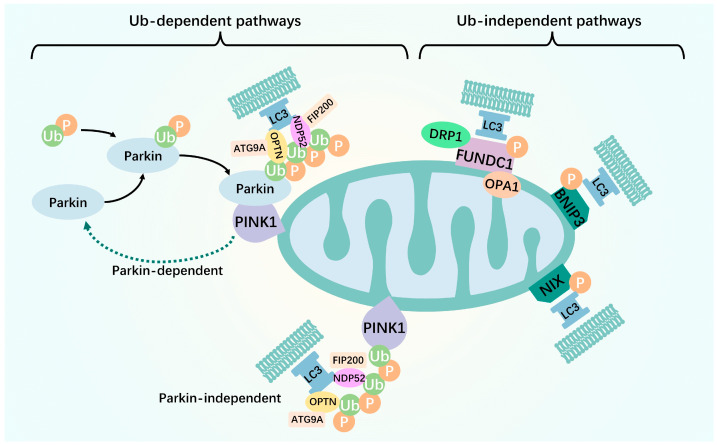
Two pathways of mitophagy. The ubiquitin-dependent mitophagy pathway is mainly mediated by the ubiquitination of mitochondrial surface proteins. Upon mitochondrial membrane potential disruption, the pathway of PINK1 into the inner mitochondrial membrane is blocked. This leads to steady PINK1 accumulation in the outer mitochondrial membrane, resulting in the recruitment of Parkin to the damaged mitochondria. Parkin then acts as an E3 ubiquitin ligase that binds ubiquitin molecules to the substrate protein, ultimately leading to the recognition and degradation of ubiquitin molecules by the ubiquitin-labeled proteasome. Recruitment of the mitophagy receptor proteins, OPTN and NDP52, to ubiquitinated mitochondria occurs simultaneously via their ubiquitin-binding domains. Subsequently, light chain 3 is recruited to damaged mitochondria allowing OPTN to initiate autophagosomes via LC3 interaction region-mediated initiation. Unlike PINK1–Parkin-mediated mitophagy ubiquitination, many proteins contain LIR structural domains on the outer or inner mitochondrial membrane, which are mitophagy receptors. They directly bind to the essential autophagy protein, LC3, without ubiquitination, thereby initiating mitophagy and selectively destroying damaged mitochondria. Green arrows: promotion effect.

**Figure 2 biomolecules-14-01270-f002:**
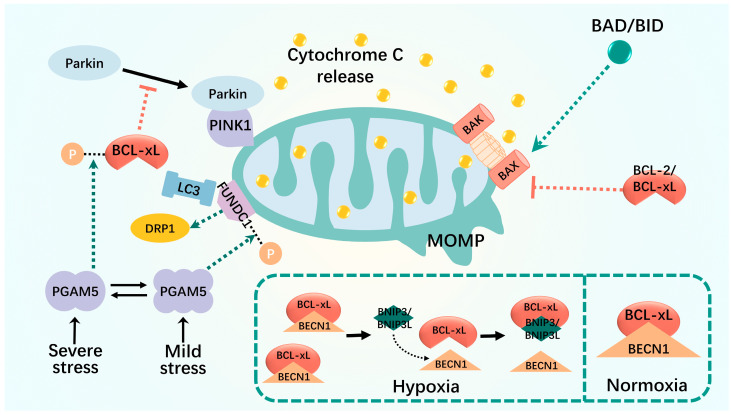
Mitophagy interacts with apoptosis. MOMP is altered in severely damaged mitochondria, which results in translocation of cytochrome c to the cytoplasm. MOMP is regulated by the BCL-2 family of proteins, including the anti-apoptotic BCL-2 and BCL-xL, the pro-apoptotic BAX and BAK, and the BH3-only proteins. The relative amount of intracellular BECLIN1 and BCL-2 (or other BCL-2 family members) interacting complexes controls the threshold for the transition from cell homeostasis to cell death. Some BCL2-interacting proteins, such as FUNDC1 and BNIP3, disrupt the interaction between BECN1 and the anti-apoptotic BCL2 family and promote the positive regulation of mitophagy. Black dashed lines: phosphorylation; green dashed arrows: promotion; red dashed arrows: inhibition.

**Figure 3 biomolecules-14-01270-f003:**
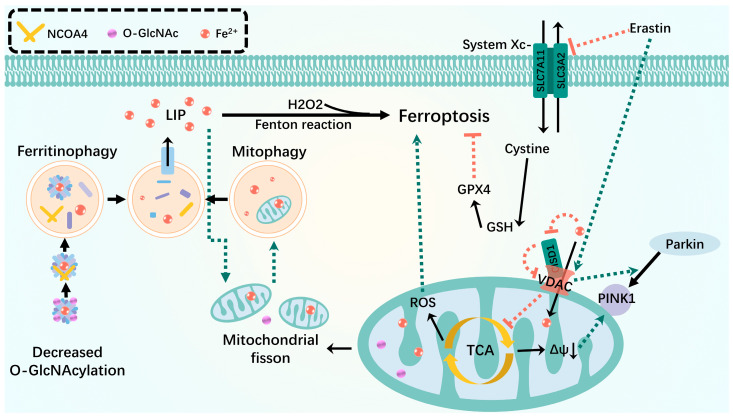
Mitophagy interacts with ferroptosis. Mitophagy isolates iron in autophagosomes during mild stress or early stages of iron overload. Inhibition of O-GlcN acylation promotes ferritinophagy, leading to the accumulation of unstable iron into mitochondria and resulting in mitochondrial breakage and enhanced mitophagy, providing an additional source of unstable iron. VDAC recruits Parkin from the cytoplasm to defective mitochondria and allows iron and metabolites to pass through the outer membrane. Erastin-induced opening of VDAC mediates mitochondrial iron uptake that accelerates ferroptosis. When the OMM protein CISD is oxidized, it oxidizes VDAC in a redox-dependent manner in the cell, closing the pore and potentially disrupting the flow of metabolites through VDAC, leading to inhibition of the mitochondrial TCA cycle and triggering an increase in mitophagy-dependent ROS, leading to ferroptosis. Green dashed arrows: promotion; red dashed arrows: inhibition.

**Figure 4 biomolecules-14-01270-f004:**
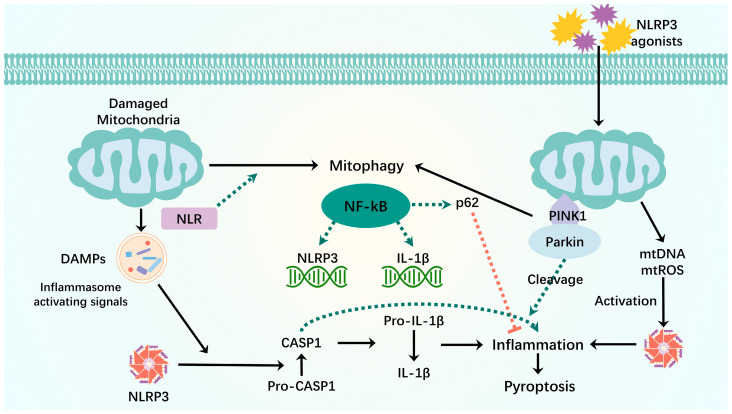
Mitophagy interacts with pyroptosis. When mitochondria are damaged, mitochondrial components are recognized as DAMPs by the cytoplasmic NLR when they are released into the cytoplasm. NLRP3-inflammatory vesicles process inactivated procaspase-1 into active caspase-1, which will cleave pro-inflammatory IL-1β into mature IL-1β, leading to inflammation and premature cell death. The binding of LIR motifs of the NLR to LC3 induces mitophagy. NF-kB activates NLRP3-inflammatory vesicles by inducing pro-IL-1β and NLRP3 expression. NF-κB induces delayed accumulation of the autophagy receptor p62/SQSTM1. External NLRP3 activation stimuli trigger mitochondrial damage and lead to direct release of mtDNA and mtROS, which activate NLRP3-inflammatory vesicles. Damaged mitochondria undergo Parkin-dependent ubiquitin binding and are specifically recognized by p62, which induces their mitophagy clearance. Green dashed arrows: promotion; red dashed arrows: inhibition.

**Table 1 biomolecules-14-01270-t001:** Comparing the similarities and differences between the ub-dependent and ub-independent pathways of mitophagy.

	Ub-Dependent Pathways	Ub-Independent Pathways
Mechanism of labeling	Recognition of ubiquitination markers and specific receptors with LIR, such as OPTN, NDP52, etc.	Directly through specific receptors (such as FUNDC1, NIX and BNIP3) interact with autophagosome.
Mainly involved in protein	PINK1 and Parkin.	DRP1, OPA1, etc.
The activation conditions	Usually activated in response to mitochondrial damage or abnormal function (e.g., loss of membrane potential).	Usually activated under specific physiological or pathological conditions (e.g., erythrocyte maturation, hypoxia).
Functional goals	Remove damaged or unwanted mitochondria to maintain cell health and function.	
Core process	Formation of autophagosomes, fusion of autophagosomes with lysosomes, and eventual degradation of mitochondria.	

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
