# Peer review of "Mitophagy in Cell Death Regulation: Insights into Mechanisms and Disease Implications"

_biomolecules, 2024, doi:10.3390/biom14101270_

Round 1
Reviewer 1 Report
Comments and Suggestions for Authors
Comments to authors
In the present study, the authors discuss the relationship between mitophagy and cell death. They first explain two types of mechanism of mitophagy, namely ubiquitin-mediated and receptor-mediated mitophagy. Next, they mentioned three main forms of cell death; apoptosis, ferroptosis, and pyroptosis. In the last chapter, they describe how mitophagy and cell death are connected. However, in the abstract the authors mention that the review offers potential directions for therapeutic applications, yet this was not further discussed in the review, reducing its actual relevance and contextualization to physiological and disease conditions. Additionally, the manuscript suffers from broad generalization without providing specific details or context from the cited work.
Major points:
- The authors should elaborate more about clinical relevance and/or disease-relevant cases and applications for the link between cell death and mitophagy. Given the relevance of this point an additional paragraph on this topic would be needed.
- Related to this point, the authors should also elaborate more in the discussion to provide perspectives on ongoing developments in terms of drugs, natural molecules or other interventions aimed at modulating mitophagy to reduce or prevent cell death when unwanted, or favor it like in the case of certain cancers.
- Throughout the text, the sentence structure requires some corrections for clarity and coherence. Some typos also need to be corrected.
- In Section 2.1, the authors describe parkin-independent mitophagy. It would be important to give context ad examples of when this process occurs (which cell types, tissues, or models in which this was observed), adding relevant references. Similarly, the other sections also suffer from broad generalization without providing more specific details or context, which is critical in describing the mechanisms of this review. This needs to be addressed in order to improve the solidity of the review.
- In the paragraph on pyropoptosis, major players are mtDNA release from mitochondria and inflammation. Recent papers have shown a link between mtDNA release, cGAS/STING and inflammatory/immune pathways and the relevance of boosting mitophagy to reduce that. The authors should include recent developments in this direction.
Minor points:
- No references to the figures were found in the text, please add them where needed
- Figures are difficult to understand.
- Not clear if the different metabolic pathways in ferroptosis are independent of each other resulting to ferroptosis or if these metabolic pathways are connected and are all involved in ferroptosis.
- In figure 1, the hypoxia related modulation of apoptosis/mitophagy proteins is not very clear, please redraw in a way that is clearer to understand.
- In figure 2, please label what the different colored circles indicate (iron, etc) to increase readability of the figure
Comments on the Quality of English Language
Throughout the text, the sentence structure requires some corrections for clarity and coherence, otherwise it makes it hard to follow in a clear manner. Some typos also need to be corrected, there are a few scattered throughout the manuscript
Author Response
Please see the attachment.
The manuscript has been sent to English editing through the "editage" service. The English editing certificate is available on request from the Editorial Office.

Reviewer 2 Report
Comments and Suggestions for Authors
General comments
The manuscript summarizes current knowledge on mitophagy and its association with the different types of cell death. The topic is without doubt of importance in basic biology and in pathology. However, the text is excessively dense and detailed for a non-specialist reader.
I have two major concerns about the manuscript. Firstly, it has been recently published good reviews focused exactly in the same aspects addressed in this manuscript (see 1!; 2); and, second, the manuscript gives excessaive attention to the different forms of cell death (already reviewed in many articles) while forgetting the possible implication of mitophagy in other degenerative processes such as cell senescence.
Detailed comments
The manuscript requires schemes summarizing the two degenerative cascades leading to mitophagy. Also, a selection of micrographs illustrating the process would be appreciated.
Reference
Mitophagy-related regulated cell death: molecular mechanisms and disease implications.
Yang M, Wei X, Yi X, Jiang DS. Cell Death Dis. 2024;15:505. doi: 10.1038/s41419-024-06804-5.
Cardiovascular disease: Mitochondrial dynamics and mitophagy crosstalk mechanisms with novel programmed cell death and macrophage polarisation.
Liu D, Qin H, Gao Y, Sun M, Wang M. Pharmacol Res. 2024;206:107258. doi: 10.1016/j.phrs.2024.107258.
Round 2
Reviewer 1 Report
Comments and Suggestions for Authors
I read through the review, and they added a new paragraph about disease relevant cases of mitophagy. I am a bit confused about their structure, since they first have a chapter of disease-relevant cases and then another chapter of targeted therapy for mitophay. Yet in the disease-relevant chapter they also mention some therapeutic strategies, which (for me) makes the structure of chapters a bit more messy/uncoherent.
The authors have performed some adjustments to the original manuscript. They rewrote specific sections that I had specifically mentioned that needed more context. However, I commented that they had to correct some sentence structure, yet I can not find any changes.
I am still confused about their structure, since they first have a chapter of disease-relevant cases and then another chapter of targeted therapy for mitophagy. Yet in the disease-relevant chapter they also mention some therapeutic strategies, which makes the structure of chapters still uncoherent. Additionally, no perspectives on future knowledge gaps to still be filled and possible therapeutic potential of these mechanisms is mentioned in the conclusions, which would usually be an important part.
Therefore, I would ask to further improve on the structure of the chapters as also initially suggested.
Reviewer 2 Report
Comments and Suggestions for Authors
The authors have addressed all my comments. Hoever I still think that the space devoted to reviewing the different types of cell death is excessive. Additionally, I do not understand the reason for omitting the reference to a very recent review on the same topic that may be useful to readers.
